# Soluble CD14 Induces Pro-inflammatory Cytokines in Rheumatoid Arthritis Fibroblast-Like Synovial Cells via Toll-Like Receptor 4

**DOI:** 10.3390/cells9071689

**Published:** 2020-07-14

**Authors:** Yoshihide Ichise, Jun Saegusa, Shino Tanaka-Natsui, Ikuko Naka, Shinya Hayashi, Ryosuke Kuroda, Akio Morinobu

**Affiliations:** 1Department of Rheumatology and Clinical Immunology, Kobe University Graduate School of Medicine, Kobe 650-0017, Japan; yichise@med.kobe-u.ac.jp (Y.I.); stanaka@med.kobe-u.ac.jp (S.T.-N.); iknaka@med.kobe-u.ac.jp (I.N.); morinobu@med.kobe-u.ac.jp (A.M.); 2Department of Clinical Laboratory, Kobe University Hospital, Kobe 650-0017, Japan; 3Department of Orthopedic Surgery, Kobe University Graduate School of Medicine, Kobe 650-0017, Japan; shayashi@med.kobe-u.ac.jp (S.H.); kurodar@med.kobe-u.ac.jp (R.K.)

**Keywords:** fibroblast-like synoviocytes, rheumatoid arthritis, Soluble CD14, TLR-4, IL-6, TNF-α, IL-17A, endogenous activator

## Abstract

Objectives: Synovial fluids of rheumatoid arthritis (RA) patients commonly contain high concentrations of soluble CD14 (sCD14). To investigate its potential role in RA pathogenesis, we tested whether sCD14 binding transmits a signal to fibroblast-like synoviocytes from RA patients (RA-FLS). Methods: The induction of pro-inflammatory cytokines, chemokines, and mediators by sCD14 stimulation of RA-FLS was quantified by real-time PCR and ELISA. Cell proliferation was assessed by the BrdU assay. LPS-RS, a Toll-like receptor 4 (TLR-4) antagonist, was used to block TLR-4 signaling. Results: Soluble CD14 induced the expression of IL-6 mRNA and secretion of the protein. The expression of other pro-inflammatory cytokines and mediators, such as TNF-α, IL-8, intercellular adhesion molecule-1 (ICAM-1), MMP-3, and RANK ligand (RANKL), was also induced by sCD14. In addition, sCD14 stimulation promoted RA-FLS proliferation. LPS-RS abolished IL-6, IL-8, and ICAM-1 mRNA induction by sCD14 in RA-FLS. On the other hand, TNF-α and IL-17A increased TLR-4 expression by RA-FLS and amplified their sCD14-induced IL-6 expression. Conclusions: Soluble CD14 transmits inflammatory signals to RA-FLS via TLR-4. The effects of sCD14 may be augmented in inflammatory milieu. Our results suggest that sCD14 is involved in the pathogenesis of RA and may be a novel therapeutic target.

## 1. Introduction

RA is characterized by chronic synovial inflammation and hyperplastic synovial tissue, which subsequently cause cartilage and bone destruction [1]. Fibroblast-like synoviocytes (FLS) in RA are a key component of this invasive synovium. They have dual characteristics as “passive responders” but also “imprinted aggressors”, i.e., the presence of activated FLS reflects a proinflammatory milieu, but on the other hand, they also act as primary promoters of inflammation, persistently synthesizing cytokines, chemokines, MMP, and adhesion molecules. This dual behavior suggests that FLS-directed therapies could become a complementary approach to the use of biological compounds that suppress the synthesis of key inflammatory cytokines such as TNF-α [2,3].

CD14 was first characterized as a membrane-associated glycosylphosphatidylinositol (GPI)-anchored protein and a cell surface differentiation marker present on monocytes, macrophages, dendritic cells, and neutrophils, where it acts as a receptor for lipopolysaccharide (LPS). Because CD14 lacks a transmembrane domain, it is unable to initiate signaling responses by itself. LPS bound to CD14 is transferred to the Toll-like receptor-4 (TLR-4)-myeloid differentiation factor 2 (MD-2) complex, where it subsequently delivers intracellular signals [4,5,6].

However, in addition to membrane-bound CD14 (mCD14), there is also a soluble form of this molecule (sCD14) [7]. Higher sCD14 concentrations are found in serum, synovial fluid (SF), and urine of RA patients than are present in OA patients or healthy controls. Moreover, earlier studies indicated that serum and urine sCD14 levels reflected RA disease activity [8,9,10]. However, the roles of sCD14 in the pathogenesis of RA are still not fully understood. Taking these results together, we hypothesized that sCD14 transduces and activates intracellular signaling in fibroblast-like synoviocytes from RA patients (RA-FLS).

In the present study, we document that sCD14 induces the expression of pro-inflammatory cytokines, chemokines, and mediators in RA-FLS via TLR-4 and promotes the proliferation of these cells. These results collectively suggest that sCD14 has important roles in RA pathogenesis and might therefore be a potential target for treatment of RA.

## 2. Materials and Methods

### 2.1. Reagents

Recombinant human CD14 was purchased from PeproTech (Rocky Hill, NJ, USA), and recombinant human TNF-α and IL-17A were purchased from R&D Systems (Minneapolis, MN, USA). Ultrapure lipopolysaccharide from *R. sphaeroides* (LPS-RS) was purchased from InvivoGen (Toulouse, France).

### 2.2. FLS and Cell Culture

Human studies were approved by the Ethics Committees of Kobe University Hospital and conducted in accordance with the Declaration of Helsinki. RA synovia were obtained after informed consent from RA patients undergoing joint-replacement surgery or synovectomy. The patients fulfilled the American College of Rheumatology 1987 criteria [11]. FLS were isolated from RA synovium by enzyme treatment, as described previously [12]. The isolated RA-FLS were cultured in Dulbecco’s modified Eagle’s medium (DMEM) supplemented with 10% fetal bovine serum (FBS) (GIBCO BRL, Palo Alto, CA, USA), 1% penicillin-streptomycin (Lonza Walkersville Inc., Walkersville, MD, USA), and 2 mM L-glutamine (GIBCO BRL). RA-FLS were used between passages 3–6 and maintained as previously described [12].

### 2.3. Reverse Transcription Quantitative PCR (RT-qPCR)

RA-FLS were seeded into 6-well plates (1 × 10^5^ cells/mL) and incubated overnight with medium containing 10% FBS. Thereafter, they were stimulated with cytokines for a set time. Total RNA was isolated by using RNeasy (Qiagen, Hilden, Germany), and 1 μg of total RNA was reverse transcribed, using QuantiTect reverse-transcription kits (Qiagen). Quantitative real-time PCR was performed by using a QuantiTect SYBR Green PCR Kit (Qiagen) and an ABI Prism 9900 instrument (Applied Biosystems, Foster City, CA, USA), according to the manufacturer’s instructions. The IL-6, TNF-α, and RANK ligand (RANKL) primer pairs were from Qiagen, and primer sequences are summarized in Table 1. The mRNA levels were normalized to glyceraldehyde-3-phosphate dehydrogenase (GAPDH; QT01192646, Qiagen).

### 2.4. ELISA

RA-FLS were seeded into 6-well plates (1 × 10^5^ cells/mL) with medium containing 10% FBS and incubated overnight. Thereafter, cells were cultured with or without sCD14 (500 ng/mL) for 24 h, followed by quantification of IL-6 in supernatants by ELISA kits for human IL-6 (R&D Systems), as per the manufacturer’s protocol. Stimulation with sCD14 was also carried out in the presence of 1 μg/mL of polymyxin B (Sigma-Aldrich, St. Louis, MO, USA).

### 2.5. Cell Proliferation Assays

RA-FLS were seeded into a 96-well plate (1 × 10^4^ cells/well) and cultured with or without sCD14 (500 or 2000 ng/mL) for 24 h. Thereafter, proliferation was determined by using a cell proliferation ELISA (BrdU; Roche, Basel, Switzerland), following the manufacturer’s instructions, and by measuring optical densities at 450 nm with a microplate reader (Bio-Rad, Hercules, CA, USA).

### 2.6. Inhibition of TLR-4

To examine the role of TLR-4 in sCD14 signaling pathway, LPS-RS Ultrapure (InvivoGen), an TLR-4 antagonist, was used to block its function. RA-FLS were seeded into 6-well plates (1 × 10^5^ cells/mL) and incubated overnight, after which they were cultured with sCD14 (500 ng/mL) in the absence or presence of LPS-RS (2 μg/mL, 1 h pretreatment) for 3 h. To determine the effect of the inhibitor, after the culture period, relative levels of IL-6, IL-8, and intercellular adhesion molecule-1 (ICAM-1) mRNA in the cells were measured by real-time PCR.

### 2.7. Statistical Analysis

Results were expressed as the mean ± SEM. Statistical analysis was performed by Student’s *t*-test or one-way analysis of variance, and Tukey’s multiple comparison testing. The *p*-values < 0.05 were considered statistically significant.

## 3. Results

### 3.1. Soluble CD14 Induces the Synthesis of IL-6 in RA-FLS

IL-6 plays a key role in the development of RA [13]. We first investigated the effect of sCD14 on the expression of IL-6 mRNA by RA-FLS. We found that IL-6 mRNA expression was induced by sCD14 at a concentration of 500 ng/mL (Figure 1A) and reached a maximum after 3 h (Figure 1B). We then examined the production of IL-6 protein. ELISA clearly showed that sCD14 induced the production of IL-6 protein by RA-FLS (Figure 1C). These results suggest that sCD14 facilitates synthesis of IL-6 by RA-FLS.

### 3.2. Soluble CD14 Induces the Expression of Pro-Inflammatory Cytokines, Chemokines, and Mediators by RA-FLS

Previous studies showed that RA-FLS produce a variety of cytokines and molecules that modulate growth, inflammation, angiogenesis, and cell recruitment [3]. To further elucidate the effect of sCD14 on RA-FLS, we assessed expression of other inflammatory mediators, in addition to IL-6. We found that sCD14 increased the expression of IL-8, ICAM-1, IL-1β, TNF-α, GM-CSF, CCL5, CXCL10, MMP-3, RANKL, and COX-2 mRNA in RA-FLS (Figure 2). These results show that sCD14 induces the expression of several different cytokines, chemokines, and mediators by RA-FLS and suggest that sCD14 could be involved in RA pathogenesis through promoting inflammation, hyperplasia, neoangiogenesis, local infiltration of immune cells, osteoclastogenesis, and matrix destruction.

### 3.3. High Concentrations of sCD14 Promote the Proliferation of RA-FLS

Active FLS proliferation in RA contributes to pannus formation [14]. We therefore studied the effect of sCD14 on proliferation of RA-FLS. We showed that relatively high concentrations of sCD14 promote the proliferation of these cells (Figure 3). These results suggest that sCD14 could be involved in stimulating synovial hyperplasia.

### 3.4. The TLR-4 Antagonist LPS-RS Abolishes sCD14-Induced IL-6, IL-8, and ICAM-1 Expression by RA-FLS

In view of the signaling mechanisms of the CD14 receptor, we hypothesized that sCD14 would transmit signals through TLR-4. We used LPS-RS, an antagonist of TLR-4, to block TLR-4 signaling. We found that pretreatment with LPS-RS almost completely inhibited the induction of IL-6, IL-8, ICAM-1 mRNA in sCD14-stimulated RA-FLS (Figure 4). Pretreatment with LPS-RS also inhibited IL-1β, TNF-α, GM-CSF, CCL5, CXCL10, MMP-3 and COX-2 mRNA expression in RA-FLS with sCD14 stimulation for 3 h (data not shown). These results show that sCD14 induces expression of these mRNAs in RA-FLS via TLR-4.

### 3.5. Addition of TNF-α or IL-17A Augments the Response of RA-FLS to sCD14

TNF-α and IL-17A play important roles in the activation of RA-FLS. Previous studies showed that both cytokines induce the production of IL-6 and promote the expression of TLR-4 in RA-FLS [15,16,17]. We evaluated the effects of TNF-α and IL-17A on TLR-4 expression in RA-FLS and showed that both of these cytokines significantly increase TLR-4 mRNA expression in RA-FLS (Figure 5A,B).

We next investigated whether treatment with TNF-α or IL-17A, together with sCD14, would augment the responses of RA-FLS. We incubated RA-FLS with either medium (control), sCD14, TNF-α or IL-17A, or a combination of sCD14 and TNF-α or IL-17A. We found that addition of TNF-α or IL-17A enhanced IL-6 expression in sCD14-stimulated RA-FLS (Figure 5C,D). These results suggest that TNF-α or IL-17A both augment the response of RA-FLS to sCD14.

## 4. Discussion

In the present study, we showed that sCD14 induced inflammation in RA-FLS via TLR-4 and also promoted the proliferation of these cells. We found that the effect of sCD14 on RA-FLS was augmented in the presence of TNF-α or IL-17A. To the best of our knowledge, this is the first report to show an important role of sCD14 as a pro-inflammatory endogenous activator in RA.

We found that sCD14 is capable of activating RA-FLS in the absence of LPS. It was previously reported that sCD14, together with a low concentration of LPS, augmented ICAM-1 expression in RA-FLS [8]. In the present study, we found that sCD14 by itself was able to increase the expression of pro-inflammatory cytokines, chemokines, and mediators in RA-FLS, as well as promote their proliferation. Concentrations of sCD14 in SF from RA patients are 2800 (2100–3300) ng/mL, emphasizing that we used relatively low concentrations of sCD14 in this study [10], but still saw strong effects. Adding polymyxin B during sCD14 stimulation had no effect on IL-6 secretion, confirming that the activity measured was not due to LPS contamination. Active FLS proliferation is of pivotal importance for pannus formation in RA [14,18]. We found that sCD14 has an effect on proliferation of RA-FLS (Figure 3) at the concentrations of 2.000 ng/mL, although the effect was relatively weak. No effect was observed on proliferation of RA-FLS by sCD14 at the concentrations of 500 ng/mL, which showed increased expressions of cytokines and chemokines (Figure 1 and Figure 2). To evaluate the role of sCD14 on cell migration, we performed a monolayer wound scratch assay. No effect was observed in the assay on sCD14-stimulated RA-FLS, even at the concentration of 2000 ng/mL (data not shown). Our results suggest that sCD14 acts on RA-FLS independently of LPS and may be involved in the pathogenesis of RA through inflammation, angiogenesis, cell recruitment, osteoclastogenesis, degrading the extracellular matrix, and encouraging synovial hyperplasia.

It has been reported that sCD14 is generated by several different mechanisms, including de novo synthesis, direct proteolytic cleavage from the cell surface, and exocytosis [19]. Soluble CD14 is produced by HepG2 hepatoma cells and peripheral blood mononuclear cells on IL-6 stimulation [10,20], suggesting that a positive feedback loop could exist whereby sCD14 causes RA-FLS inflammation, and the IL-6 produced by activated RA-FLS promotes further sCD14 production by hepatocytes and monocytes. A molecular analysis showed that proteolytic cleavage of mCD14 from synovial macrophages was important for sCD14 production [8]. Because exocytosis is a reasonable hypothesized mechanism responsible for the elevated amounts of sCD14 present in human macrophage supernatants, it was suggested that sCD14 acts as a danger-associated molecular pattern (DAMP) [19]. Further investigations are needed to understand the mechanisms responsible for sCD14 elevation in the SF of RA patients.

We report that TLR-4 is a key receptor in the sCD14 signaling pathway of RA-FLS. These cells have been reported to express TLRs. TLR-4 appears to be more important than other TLRs in RA [21]. Soluble CD14 activated RA-FLS (Figure 1 and Figure 2) but not FLS from OA patients (OA-FLS) [22]. In addition, sCD14 has been shown to activate macrophages [19] and oral epithelial cells [23], but not human coronary artery endothelial cells (HCAEC) [24]. The mechanisms behind this discrepancy in the different observed cellular responses to sCD14 are not well-understood. Levels of expression of TLR-4 in RA-FLS are higher than in OA-FLS [16,17]. We showed that blocking TLR-4 abolished the production of cytokines by sCD14-stimulated RA-FLS. Macrophages and oral epithelial cells also express TLR-4 on the cell surface [25,26], while HCAECs do not [27,28]. Thus, the level of expression of TLR-4 may be important for the susceptibility to sCD14 stimulation.

Some endogenous ligands, such as heat-shock proteins, high-mobility group B-1, S100 proteins, and tenascin-C, have been reported to be involved in TLR-4 signaling [2,29,30,31]. They are known as DAMPs, endogenous proinflammatory molecules generated upon tissue injury. However, little is known about the specific ligands that activate RA-FLS via TLR-4. S100A8 has been reported to be more highly expressed in RA synovial tissue (ST) and SF than in OA, and it induced higher levels of IL-6 production by RA-FLS than by OA-FLS, also via TLR-4. However, S100A8 stimulation did not induce the production of TNF-α, IL-1β, and other proinflammatory cytokines by RA-FLS. Moreover, TLR-4 inhibition did not completely suppress IL-6 production by S100A8-stimulated RA-FLS, suggesting that other receptors, such as RAGE, may also be involved [31]. Tenascin-C has also been shown to be increased in ST and SF of RA. Interestingly, fibrinogen-like globe (FBG), a domain of tenascin-C, induced IL-6 and IL-8 synthesis in human macrophages, but did not induce IL-8 in RA-FLS [2]. In our study, sCD14 induced the expression of both IL-6 and IL-8 in RA-FLS. Soluble CD14 may differ from other DAMPs with regard to mechanisms exhibiting synergistic effects with LPS to induce production of a large variety of cytokines by RA-FLS. Taken together, our results suggest that sCD14 acts as an important endogenous ligand for RA-FLS via TLR-4.

We showed that TNF-α and IL-17A augmented the expression of IL-6 in sCD14-stimulated RA-FLS. Excessive amounts of these cytokines were observed in SF of RA patients. These cytokines play critical roles in the pathogenesis of RA, for example, by inducing IL-6 production by RA-FLS [15]. In addition, our results are consistent with previous reports showing that these cytokines increased the expression of TLR-4 in RA-FLS [16,17]. TNF-α and IL-17A may contribute to the pathogenesis of RA by inducing cytokine production and increasing the sensitivity of RA-FLS to sCD14.

There remain many unmet needs for the optimal treatment of RA. Biological disease-modifying antirheumatic drugs, such as anti-TNF-α antibodies, have shown considerable efficacy in clinical practice, but many patients continue to have active disease [32]. TLR-4 inhibitor therapy for RA is considered one further promising candidate [33], and several TLR-4 inhibitors have demonstrated some therapeutic effect in vitro and in vivo [34]. In fact, an anti-TLR-4 monoclonal antibody has now advanced to a phase II clinical trial [35]. Further understanding of endogenous TLR-4 ligands in RA pathogenesis will be essential for appropriate and effective use these of new therapeutic agents. It was reported that relatively high concentrations of sCD14 are present in the serum of RA patients after TNF-α inhibitor therapy (about 1900 ng/mL) [10]. Therefore, TLR-4 inhibitor therapy could provide a novel complementary treatment, especially in RA patients who have an inadequate response to TNF-α inhibitors and high levels of sCD14 in SF or serum.

In the present study, we have found indications of potential roles of sCD14 for the pathogenesis of RA. Amplification of the responses of RA-FLS to sCD14 by inflammatory cytokines may reflect the complex inflammatory milieu in RA. For realizing the optimal use of TLR-4 inhibitors, it will be important to clarify the mechanisms by which endogenous molecules, such as sCD14, activate RA-FLS. Taken together, our results may point the way to a novel therapeutic strategy involving the blockade of sCD14/TLR-4 signaling pathways in RA.

## Figures and Tables

**Figure 1 cells-09-01689-f001:**
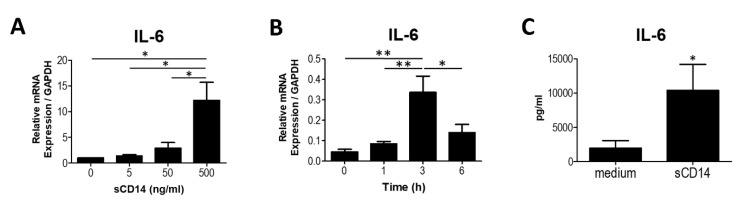
Soluble CD14 induces the expression and secretion of IL-6 by fibroblast-like synoviocytes from RA patients (RA-FLS). (**A**) RA-FLS were cultured with sCD14 (0–500 ng/mL) for 6 h. After culture, relative expression of IL-6 mRNA was measured by real-time PCR and levels normalized to GAPDH mRNA. Data are shown as means ± SEM in five RA-FLS. (**B**) RA-FLS were cultured with sCD14 (500 ng/mL) for 1, 3, and 6 h. After culture, relative expression of IL-6 mRNA was measured by real-time PCR and levels normalized to GAPDH mRNA. Data are shown as means ± SEM in five RA-FLS. Statistical significance was analyzed by Tukey’s Multiple Comparison Test (* *p* < 0.05, ** *p* < 0.01). (**C**) RA-FLS were cultured with sCD14 (500 ng/mL) for 24 h. IL-6 protein in supernatant was measured by ELISA. Data are shown as means ± SEM in three RA-FLS. Statistical significance was analyzed by unpaired *t*-test (* *p* < 0.05).

**Figure 2 cells-09-01689-f002:**
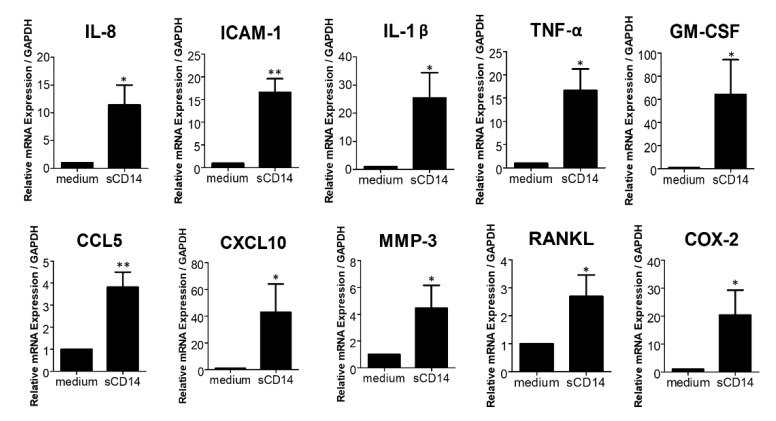
Soluble CD14 induces the expression of proinflammatory cytokines/chemokines by RA-FLS. RA-FLS were cultured with or without sCD14 (500 ng/mL) for 3 and 6 h. Thereafter, relative expression levels of IL-8, ICAM-1, IL-1β, TNF-α, GM-CSF, CCL5, CXCL10, MMP-3, RANKL, and COX-2 mRNA were measured by real-time PCR and were normalized to GAPDH mRNA. The mRNA expression of RANKL was exhibited at 6 h, while the mRNA expressions of the others were exhibited at 3 h. Representative data are shown for the two incubation periods. Data are shown as means ± SEM in five RA-FLS. Statistical significance was analyzed by unpaired *t*-test (* *p* < 0.05, ** *p* < 0.01).

**Figure 3 cells-09-01689-f003:**
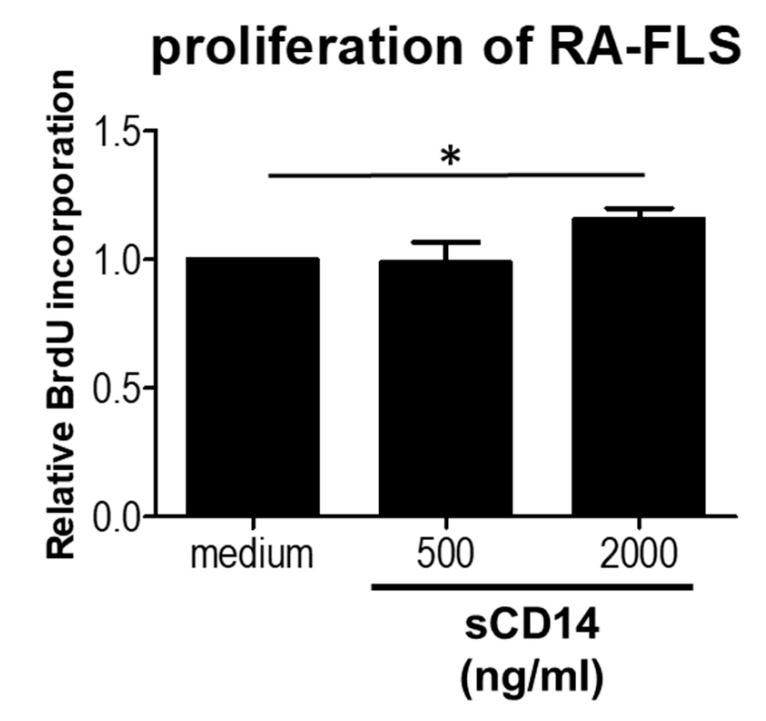
High concentrations of sCD14 induce the proliferation of RA-FLS. RA-FLS were cultured with or without sCD14 (500 or 2000 ng/mL) for 24 h. Cell proliferation was determined by using the BrdU assay. Each experiment was performed in quintuplicate. Data are shown as means ± SEM in three RA-FLS. Statistical significance was analyzed by Tukey’s Multiple Comparison Test (* *p* < 0.05).

**Figure 4 cells-09-01689-f004:**
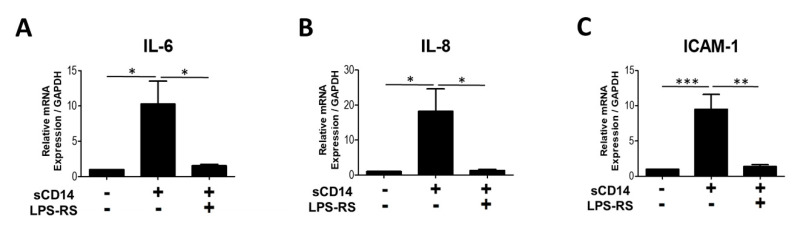
LPS-RS, an antagonist of TLR-4, abolishes sCD14-induced IL-6, IL-8, and ICAM-1 expression on RA-FLS. RA-FLS were cultured with sCD14 (500 ng/mL), in the absence or presence of LPS-RS (2 ug/mL, 1 h pretreatment), for 3 h. Relative expression levels of IL-6 (**A**), IL-8 (**B**), and ICAM-1 (**C**) mRNA were measured by real-time PCR, and their levels were normalized to GAPDH mRNA. Data are shown as means ± SEM in five RA-FLS. Statistical significance was analyzed by Tukey’s Multiple Comparison Test (* *p* < 0.05, ** *p* < 0.01, and *** *p* < 0.001).

**Figure 5 cells-09-01689-f005:**
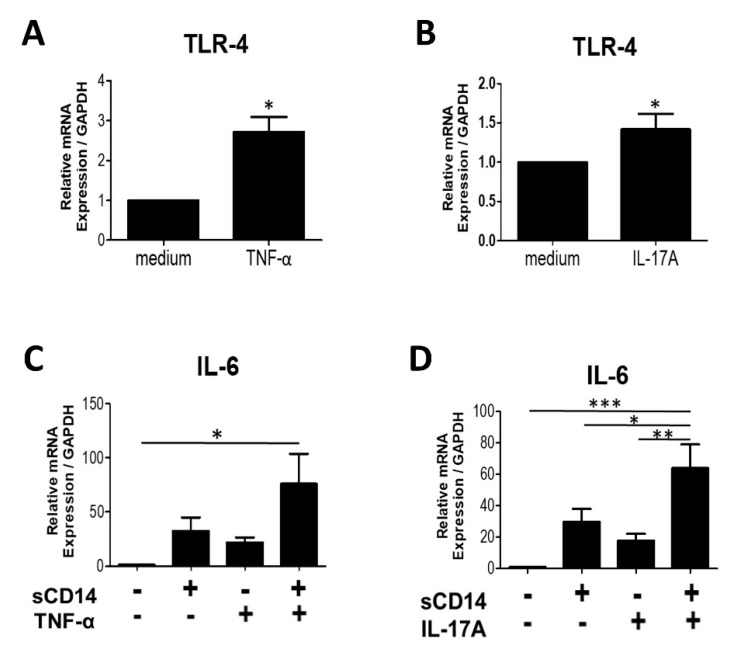
Addition of TNF-α or IL-17A augments the response of RA-FLS to sCD14. (**A**,**B**) RA-FLS were cultured with or without TNF-α (10 ng/mL) (**A**) or IL-17A (10 ng/mL) (**B**) for 3 h. Thereafter, relative expression levels of TLR-4 mRNA were measured by real-time PCR and normalized to GAPDH. Data are shown as means ± SEM in five (**A**) and nine (**B**) RA-FLS. Statistical significance was analyzed by unpaired *t*-test (* *p* < 0.05). (**C**,**D**) RA-FLS were cultured with sCD14 (500 ng/mL), TNF-α (10 ng/mL) or sCD14+TNF-α (**C**), or sCD14 (500 ng/mL), IL-17A (10 ng/mL) or sCD14+IL-17A (**D**) for 3 h. After culture, relative expression levels of IL-6 mRNA were measured by real-time PCR and normalized to GAPDH. Data are shown as means ± SEM in five (**C**) and nine (**D**) RA-FLS. Statistical significance was analyzed by Tukey’s Multiple Comparison Test (* *p* < 0.05, ** *p* < 0.01, and *** *p* < 0.001).

**Table 1 cells-09-01689-t001:** List of the sequence of gene primers.

Gene Name	Forward (5′ to 3′)	Reverse (5′ to 3′)
IL-1β	AAACAGATGAAGTGCTCCTTCCAGG	TGGAGAACACCACTTGTTGCTCCA
IL-8	ATGACTTCCAAGCTGGCCGTGGCT	TCTCAGCCCTCTTCAAAAACTTCTC
ICAM-1	ATGCCCAGACATCTGTGTCC	GGGGTCTCTATGCCCAACAA
GM-CSF	CACTGCTGCTGAGATGAATGAAA	GTCTGTAGGCAGGTCGGCTC
CCL5	TGC CTC CCA TAT TCC TCG G	CTA GCT CAT CTC CAA AGA
CXCL10	GAAATTATTCCTGCAAGCCAATTT	TCACCCTTCTTTTTCATTGTAGCA
MMP-3	TGGCATTCAGTCCCTCTATGG	AGGACAAAGCAGGATCACAGTT
COX-2	ATTGACCAGAGCAGGCAGAT	CAGGATACAGCTCCACAGCA
TLR-4	CAGAGTTGCTTTCAATGGCATC	AGACTGTAATCAAGAACCTGGAGG

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
