# Peer review of "Soluble CD14 Induces Pro-inflammatory Cytokines in Rheumatoid Arthritis Fibroblast-Like Synovial Cells via Toll-Like Receptor 4"

_cells, 2020, doi:10.3390/cells9071689_

Round 1

Reviewer 1 Report

The most important aspect of the study was the concentration of sCD14 used in their experiments (500 ng/mL), i.e., whether or not they were within the physiological range. As the authors state in the discussion, the concentration of sCD14 in the synovial fluid from RA patients was reported to be 2,100-3,300 ng/mL and in the serum of RA patients after TNF, treatment was 1,900 ng/mL.

The manuscript is well-written. Few minor points

line 94: 105

line 100: 104

line 108: 105

line 109: μg

Author Response

Thank you very much for carefully reading our manuscript. As the reviewer pointed out, we have corrected the points that the reviewers pointed out.

line 84

line 96

line 102

line 110

Reviewer 2 Report

See attached file. 

Author Response

Thank you very much for carefully reading our manuscript. We have revised the manuscript according to the reviewers' comments and have performed additional experiments. See attached file.

Reviewer 3 Report

The study of Ichise et al. examines the effects of soluble CD14 on arthritic synovial fibroblasts. For this purpose the authors assessed cytokine gene expression and proliferation of synovial fibroblasts derived from rheumatoid arthritis patients that were stimulated with sCD14. sCD14 stimulated cytokine expression via toll-like receptor 4 (TLR4) and TNFa and IL-17 amplified soluble CD14-induced cytokine expression.

The manuscript is presented in a clear and comprehensive way and the results are presented properly. In principle, the study is of interest. However, the paper would improve in terms of significance if the authors could provide more mechanistically details.

Specific comments:

  • The authors propose that sCD14 transmits the pro-inflammatory signals via TLR4. As TLR4 can signal via different downstream pathways such as NF-kB and MAPK, which pathway is involved in sCD14 signaling?
  • The study shows well that cytokine gene expression is promoted by sCD14 in RA-FLS. How are the effects in comparison to non-RA FLS such as OA-FLS?
  • The authors suppose that sCD14 promotes proliferation of FLS. However, the effects shown here are quite small and not very convincing, especially as only the 4-fold higher concentration of sCD14 exerts some effects on proliferation. Could the authors also apply other functional assays such as migration and invasion behavior of FLS to get more insights?
  • Ichise et al. speculate that TNFa and IL-17 may contribute to the pathogenesis of rheumatoid arthritis by inducing cytokine production via increasing the sensitivity of RA-FLS to sCD14. To corroborate this hypothesis the authors could include experiments stimulating FLS with sCD14 and blocking IL-17 and TNFa, respectively. 
  •  

Author Response

(The authors gave the same response as above.)

Round 2

Reviewer 2 Report

The authors have addressed my concerns. 

Author Response

Dear Reviewer:

We thank the reviewer for your critical comments and useful suggestions. We hope the manuscript is now suitable for publication in Cells

Sincerely,

Yoshihide Ichise

Reviewer 3 Report

The majority of my criticisms and queries have been answered or discussed and I now feel that the study is appropriate to be published. I just would suggest adding the new data showing that sCD14 transmits the pro-inflammatory signals via the NF-kB pathway in one of the main figures.
Furthermore, I would add a section about proliferation and migration properties of sCD14 treated FLS in the discussion.

Author Response

Dear Reviewer:

Thank you very much for carefully reading our manuscript and answers. We have revised the manuscript according to the reviewers' comments and suggestions. See attached file. We hope the manuscript is now suitable for publication in Cells.

Sincerely,

Yoshihide Ichise
